# Role of Guinea Pigs (*Cavia porcellus*) Raised as Livestock in Ecuadorian Andes as Reservoirs of Zoonotic Yeasts

**DOI:** 10.3390/ani12243449

**Published:** 2022-12-07

**Authors:** Lenys Buela, Mercy Cuenca, Jéssica Sarmiento, Diana Peláez, Ana Yolanda Mendoza, Erika Judith Cabrera, Luis Andrés Yarzábal

**Affiliations:** 1Carrera de Bioquímica y Farmacia, Unidad Académica de Salud y Bienestar, Universidad Católica de Cuenca, Av. Las Américas, Cuenca 010101, Ecuador; 2Carrera de Medicina Veterinaria, Unidad Académica de Ciencias Agropecuarias, Universidad Católica de Cuenca, Av. Las Américas, Cuenca 010101, Ecuador; 3Carrera de Odontología, Unidad Académica de Salud y Bienestar, Universidad Católica de Cuenca, Av. Las Américas, Cuenca 010101, Ecuador; 4Centro de Investigación, Innovación y Transferencia de Tecnología (CIITT), Universidad Católica de Cuenca, Ricaurte 010162, Ecuador

**Keywords:** guinea pig, *Cavia porcellus*, yeasts, antifungal resistance, zoonotic diseases, opportunistic fungi, pathogenic fungi

## Abstract

**Simple Summary:**

Guinea pigs are reared not only to be kept as pets, but also for human consumption. This happens mostly in the Andean countries (Ecuador, Bolivia, and Peru), where guinea pig meat is one important source of animal protein. In this region, animal husbandry is performed usually by small farmers, who enter into frequent and close contact with guinea pigs. This poses a potential threat to human health because these (and other domestic) animals carry opportunistic human pathogens in their tissues and organs. Using traditional microbiological procedures and molecular biology techniques, we show here that the nasal mucosa of guinea pigs may contain up to 11 species of potentially pathogenic yeasts. Several of these yeasts are resistant to compounds used to treat fungal infections, which warns against their virulence potential if acquired by humans or other animals. We suggest that more attention should be given to this situation to prevent the risk of infectious diseases caused by microbes that are transmitted from animals to humans (=zoonoses).

**Abstract:**

Guinea pigs (*Cavia porcellus*) have been reared for centuries in the Andean region for ceremonial purposes or as the main ingredient of traditional foods. The animals are kept in close proximity of households and interact closely with humans; this also occurs in western countries, where guinea pigs are considered pets. Even though it is acknowledged that domestic animals carry pathogenic yeasts in their tissues and organs that can cause human diseases, almost nothing is known in the case of guinea pigs. In this work we used traditional microbiological approaches and molecular biology techniques to isolate, identify, and characterize potentially zoonotic yeasts colonizing the nasal duct of guinea pigs raised as livestock in Southern Ecuador (Cañar Province). Our results show that 44% of the 100 animals studied were colonized in their nasal mucosa by at least eleven yeast species, belonging to eight genera: *Wickerhamomyces*, *Diutina*, *Meyerozyma*, *Candida*, *Pichia*, *Rhodotorula*, *Galactomyces*, and *Cryptococcus*. Noticeably, several isolates were insensitive toward several antifungal drugs of therapeutic use, including fluconazole, voriconazole, itraconazole, and caspofungin. Together, our results emphasize the threat posed by these potentially zoonotic yeasts to the farmers, their families, the final consumers, and, in general, to public and animal health.

## 1. Introduction

For millennia, guinea pigs (*Cavia porcellus*) have been raised in the Andean region either for ceremonial purposes or as the main ingredient of traditional foods. In countries such as Bolivia, Peru and Ecuador, these rodents are reared by small farmers for family subsistence or are mass-produced as livestock to be sold in markets and supermarkets. Guinea pigs can be raised in small numbers in artisanal facilities located in the close vicinity of households, being nurtured and treated with care; they can also be raised in huge barns under intensive feeding regimes for meat production [1,2]. On the contrary, in many countries of the western hemisphere, guinea pigs are considered easy-care pets and, thus, they are reared indoors and in intimate contact with their owners (mostly children). 

According to the Food and Agriculture Organization (FAO) of the United Nations, in 2009, the stable population of guinea pigs in the Andean region reached approximately 36 million animals for human consumption [3]. In Ecuador, more than 700.00 families depend on guinea pig production and commercialization for their subsistence [4]. Unfortunately, almost no measures of sanitary control are taken by small farmers to avoid health problems related to careless handling of these animals.

Yet, in the last 20 years, a few reports have warned about the identification of guinea pigs as being either accidental hosts or as bona fide reservoirs for human pathogens [5,6,7,8,9]. In the particular context of the Tropical Andes, guinea pigs have been shown to be infected by *Fasciola hepatica* [10], *Yersinia pestis* [11], *Trypanosoa cruzi* [12], *Campilobacter jejuni*, Shiga toxin-producing *Escherichia coli* [2,13], and methicillin-resistant *Staphylococcus aureus* [13]. Guinea pigs have also been shown to carry influenza virus antibodies in their blood [14].

Among opportunistic human pathogens, yeasts—especially *Candida* species—stand out. However, these are far from being the only worrying species; instead, new emerging fungal pathogens include members of other lesser-known genera, such as *Malassezia*, *Trichosporon*, *Rhodotorula*, and *Wickerhamomyces* [15]. Animals, either wild or domesticated, may act as natural or accidental reservoirs for these emerging pathogens, playing an essential role in their transmission to human populations [16,17]. Noticeably, it has been recently established that almost 37% of the 202 zoonotic events registered in the period of 1940–2004 were related to animals kept in captivity for food production [18]. These animals were considered as reservoirs of the pathogens in 64 of these events, and as intermediates or amplifying hosts in another 8 events.

Some domestic animals, such as pigeons and horses, have already been shown to carry opportunistic pathogenic yeasts in their tissues and organs [19,20,21]. However, in the case of guinea pigs raised for human consumption—and despite the abovementioned proximity and close interaction with humans—almost nothing is known. The absence of these kinds of studies is striking, since zoonotic fungi have been known for centuries and are considered as threats to human and animal health [17,22,23]. Thus, the initial aim of the present study was to isolate and identify *Candida* spp. species colonizing the nasal mucosa of healthy guinea pigs raised in Southern Ecuador as livestock, and to test their susceptibility against several antifungal drugs. Later on, we also aimed to detect other potentially zoonotic yeast species colonizing the same environment.

## 2. Materials and Methods

### 2.1. Sampling Area and Animals

A total of 100 guinea pigs from 10 farms in the municipality of Biblián (2°42′36″S, 78°52′48″W), Cañar (Southern Ecuadorian Andes) (Figure 1), were included in this survey, which was performed from March to November 2021.

Samples were collected indistinctly from clinically healthy adult animals. According to national regulations in Ecuador, sample collections for the diagnosis in farm animals do not require approval from an ethics committee (“Ley Orgánica de Sanidad Agropecuaria” 2017, Asamblea Nacional, República del Ecuador) [24]. Nevertheless, the farm owners were asked to sign a written consent and were present during the collection of samples.

### 2.2. Sample Collection and Isolation Procedures

Samples were collected from the nasal cavity of healthy adult guinea pigs by using sterile cotton swabs. The swabs were moistened with sterile saline solution before collecting the samples and were gently rubbed inside the nose cavity. Once finished, the swab tips were introduced into sterile glass tubes containing Trypticase soy broth (Oxoid) and stored in coolers to be transported to the laboratory. Petri dishes containing Hi-Crome Candida Differential Agar (HiMedia Laboratories, Mumbai, India) were inoculated by gently spreading the sample in one side, and then streaked with an inoculating loop to obtain single colonies. Incubation was carried out at 30 °C for up to six days, with daily observations to detect yeast colonies. All colonies with a microscopic morphology consistent with yeasts were re-streaked several times until purification and were further identified by molecular methods (see below).

Isolates were stored in distilled water at room temperature, in the dark, and were recovered from storage by plating onto Potato Dextrose Agar (Difco Laboratories, Bergen, NJ, USA) for further experimentation.

### 2.3. Antifungal Susceptibility Tests

Susceptibility to four antifungal drugs of a subgroup of 22 selected pure isolates was ascertained by the disk diffusion technique, according to the Clinical and Laboratory Standards Institute (CLSI) [25]. The assay was performed on Mueller–Hinton agar (MHA) plates, supplemented with 2% glucose and 0.5 µg ml^−1^ methylene blue dye. The antifungals tested were fluconazole (25 µg), voriconazole (1 µg), caspofungin (5 µg) (Liofilchem S.R.L., Rosetto degli Abruzzi, Italy), and itraconazole (10 µg) (Bioanalyse, Ankara, Turkey). The diameter of the zone of inhibition of yeast growth was measured and registered after 24 h at 30 °C. The *Candida albicans* ATCC 90028 strain was used as reference.

### 2.4. DNA Extraction and Molecular Identification of the Isolates

Yeast DNA extraction was performed according to the rapid-boiling method described by Silva et al. [26]. The species identification was performed by PCR amplification of the rDNA internal transcribed spacer (ITS), nucleotide sequencing of the amplicons, and further analysis. For this, forward (ITS1 5′-TCC GTA GGT GAA CCT GCG G-3′) and reverse (ITS4 5′-TCC TCC GCT TAT TGA TAT GC-3′) primers were used [27,28]. The PCR mixture contained 25 μL premix (DreamTaq Green PCR Mastermix, Thermoscientific), 19 μL deionized sterile water, 2 μL from each forward and reverse primer, and 2 μL of genomic DNA, which served as the DNA template in a final volume of 50 μL. The PCR cycling conditions were: an initial denaturation phase at 95 °C for 5 min, followed by 35 cycles of denaturation at 94 °C for 1 min, annealing at 58 °C for 1 min, and extension at 72 °C for 1 min, with a final extension phase at 72 °C for 10 min.

Amplicons were sequenced by Macrogen (Seoul, Republic of Korea) and the nucleotidic sequences were subjected to BLAST searches against fungal sequences existing in DNA databases (http://blast.ncbi.nlm.nih.gov/Blast.cgi accessed on 7 June 2022). The sequences were compared using nucleotide–nucleotide BLAST (blastn) with default settings, except that the sequences were not filtered for low complexity. Species were identified based on the highest similarity score (100%) with the reference database sequence. All nucleotide sequences were deposited in GenBank.

## 3. Results

One or several yeast colony morphotypes were retrieved from forty-four out of one hundred healthy guinea pigs (44%) reared in ten farms in the Cañar Province (Figure 2a,b).

All isolates exhibited a characteristic yeast morphology at the microscopic level (Figure 2c,d). The number of colony morphotypes retrieved in Hi-Crome Candida Differential Agar from each animal ranged from one (13 animals) to five (1 animal).

Incidentally, fewer animals were colonized in their nasal mucosa by yeasts when raised in ground pits and fed with natural fodder (five farms: A, B, C, D, and E) (Figure 3a), as compared to animals raised intensively in cages (five farms: F, G, S, M, and R) and fed with a mixture of fodder and cereal concentrates (Figure 3b).

In order to determine the diversity of yeast species colonizing the nasal duct of guinea pigs, we selected a subgroup of 30 isolates showing conspicuous morphological differences (or showing similar phenotypes but derived from different animals) at the colony level for molecular identification. The results obtained by nucleotide sequencing and BLAST analysis of the chromosomal ITS1-ITS4 region allowed us to identify these isolates as members of eleven yeast species, belonging to eight different genera: *Wickerhamomyces*, *Diutina*, *Meyerozyma*, *Candida*, *Pichia*, *Rhodotorula*, *Galactomyces*, and *Cryptococcus* (Table 1).

Several of these isolates were subsequently tested for their susceptibility to four antifungal drugs (Table 2). The results show that 15 out of 22 yeast isolates were insensitive to at least one of the antifungal drugs tested. Some isolates were insensitive to two or three of these antifungals. Itraconazole (at the dose tested) was shown to be ineffective against 14 (63.6%) of the isolates tested, followed by caspofungin and fluconazole (4/22, 18.2%), and, finally, voriconazole (3/22, 13.6%). On the other hand, three *Pichia* spp. were insensitive to three of the drugs tested (namely fluconazole, itraconazole, and voriconazole), while three *Meyerozyma* spp. were insensitive to two of them (itraconazole and caspofungin).

## 4. Discussion

Interactions between humans and farm animals are at the origin of several zoonotic diseases which may threaten the health of farmers, their families, the final consumers, and—in general—the whole community. Considering that (i) guinea pigs are among the most important sources of animal protein for human consumption in the Andean region; (ii) these rodents are frequently handled by breeders, distributors, sellers, and consumers; and (iii) almost no safety measures are taken during breeding, slaughter, sale, and preparation, a better understanding of their role as potential reservoirs of opportunistic pathogens is necessary. In the present work, our initial aim was to prospect the nasal mucosa of guinea pigs for opportunistic *Candida* spp. strains. Surprisingly, as we have shown here, at least 11 yeast species colonized the nasal mucosa of guinea pigs raised for human consumption, some of which having already been identified as opportunistic human pathogens. As far as we know, this is the first report showing that raised guinea pigs are an important reservoir of these kinds of potentially pathogenic non-*Candida* yeasts.

The presence of yeasts in the nasal mucosa of guinea pigs was not totally unexpected. In fact, it is well known that, due to its constant humidity, this particular environment is colonized by all kinds of microbes, including saprophytic or pathogenic fungi. Opportunistic yeast species have also been shown to persist outside of clinical settings, mainly in soils and plants, which can also act as natural reservoirs. This suggests that the ecology of these fungi may be more complex than previously thought [29].

Non-*albicans Candida* and other rare yeast species have emerged in recent decades as agents of serious diseases in humans, especially targeting immunocompromised individuals, children, and the elderly [30]. Among the virulence factors expressed by many of these pathogens, resistance against antifungal drugs is of utmost importance [31]. Since no specific clinical breakpoints have been established to date, by either the European Committee on Antimicrobial Susceptibility Testing (EUCAST) or the CLSI, for environmental or rare opportunistic yeasts, we decided to present here only the descriptive results (inhibition halo diameter, expressed in mm values), with no attempt to categorize the isolates according to the results of the antifungal susceptibility tests (except when the inhibition of growth was complete). Nevertheless, we also considered that the benefits and advantages of the disk diffusion tests, as tools for antifungal resistance surveillance, have been previously shown, as in the case of the ARTEMIS DISK Global Antifungal Surveillance Study [32]. In the case of fluconazole and voriconazole, disk testing is highly accurate and can be routinely performed in clinical laboratories.

*Wickerhamomyces anomalus* (formerly *Pichia anomala*) belongs to this group of non-*Candida*, non-*Cryptococcus* yeasts of clinical interest. It is considered an environmental yeast, usually found in soils, plants, and fruits, but also in the feces of bats and birds [21,33,34]. *W. anomalus* is increasingly being considered as an emerging pathogen of clinical importance, often being associated with a wide range of fungal infections, from ocular keratitis to meningitis and fatal candidemia [35]. Even though it is believed to be ubiquitous in the environment, the knowledge concerning *W. anomalus* ecology is still limited.

Our results show that more than half of the *W. anomalus* isolates tested were completely insensitive to itraconazole. From a public health perspective, this finding is of relevance since previous studies have revealed that some nosocomial *W. anomalus* isolates can, in fact, tolerate in vitro high doses of azoles, including flucytosine, itraconazole, and fluconazole [35,36,37].

*Diutina catenulata* (formerly *Candida catenulata*), an ascomycete closely related to *Saccharomyces cerevisiae* and often used for bioremediation purposes [38,39], was also present in the nasal mucosa of the healthy guinea pigs studied. This species has been previously isolated from the gut of birds [20,40,41] and the nasolacrimal duct of healthy horses [18]. More importantly, a few reports have documented human infections with *D. catenulata* and increasingly recognize its importance as an emerging pathogen [42].

*D. catenulata* animal isolates produce and secrete potent hydrolases (e.g., proteases and phospholipases), as previously shown by Brilhante et al. [19] and Rhimi et al. [43]. These enzymes are considered as virulence factors, which play fundamental roles in the pathogenesis of yeast infections [44]. In addition, *D. catenulata* isolates generally exhibit antifungal sensitivity, particularly to azoles and echinocandins [20,42,45,46]. However, it has been shown that some nosocomial isolates can tolerate high concentrations of fluconazole or caspofungin [47]. Thus, it is important to explore their susceptibility to these antifungals. In the present report, we show that one *D. catenulata* isolate was insensitive to caspofungin, while another two were slightly inhibited by this antifungal drug. This opens the question regarding their virulence potential if spread to human hosts, since echinocandins represent the first choice of treatment against invasive candidiasis in patients [48].

On the other hand, three fungal isolates were identified as *Meyerozyma caribbica* (anamorph *Candida fermentati*), and a fourth as *Meyerozyma carpophila* (formerly *Candida carpophila*). These phenotypically indistinguishable species are phylogenetically related and belong to the *M. guilliermondii* species complex [49,50]. Even though yeasts of this complex are ubiquitous in the environment and can be isolated from a variety of sources—including fermented foods, plants, and arthropods—some isolates can also cause human infections [51,52]. Furthermore, species within this complex are increasingly regarded as emerging infectious yeasts of the non-*albicans Candida* species group, which are considered as opportunistic pathogens in immunocompromised patients and are responsible for 1–5% of nosocomial bloodstream infections worldwide [23].

In a recent report, Chaves and coworkers isolated *M. caribbica* and *M. guillermondii* strains from patients suffering from bloodstream infections in an oncology reference center in Brazil [52]. The isolates were poorly susceptible to antifungals and exhibited high minimum inhibitory concentrations (MICs) for fluconazole and echinocandins. Strikingly, all four tested *Meyerozyma* spp. isolates studied here were insensitive to itraconazole, and three of them to caspofungin (an echinocandin). Again, these results raise concerns about their potential as virulent zoonotic pathogens.

We also identified five isolates belonging to *Candida railenensis* or *C. parapsilosis*. Both species belong to the non-*albicans Candida* group, some of which cause important infections in humans and whose incidence has been increasing in recent years [22]. Most of these species are generally considered commensal microorganisms, frequently found in the skin and mucosae (respiratory, genital, and gastrointestinal) of animals such as dogs, horses, and birds [53,54]. However, they are also considered emerging pathogens and must be considered with care, particularly in the case of *C. parapsilopsis* isolates, since it is the second most frequently isolated opportunistic *Candida* species [55].

The identification of an *R. glutinis* isolate is of upmost importance. This pink yeast belongs to a very heterogeneous group, commonly found in the tissues and organs of animals; it can also be easily isolated from environmental samples [56]. Although infections caused in humans by *R. glutinis* are less frequent than those caused by *R. mucilaginosa*, it is an opportunistic pathogen, which can infect the blood, the central nervous system, the eyes, and the heart, among other organs and tissues [57]. *Rhodotorula* species appear to be intrinsically resistant to fluconazole, which has been predicted to be ineffective to treat infections caused by this fungus [58]. In line with this information, the only *R. glutinis* isolate tested here was insensitive to fluconazole and itraconazole, a result that adds to its potential as a virulent zoonotic pathogen.

As said above, the presence of commensal or opportunistic pathogenic fungi in the nasal cavity of animals was far from unexpected. In fact, it is known that the nasal resident bacteria interact with the immune system of the mammal host, and this dynamic interaction favors the colonization of the mucosa by other microbes, such as *Candida* spp. and other yeasts [58,59]. In this particular environment, microbes—either bona fide commensals or opportunistic pathogens—are sheltered and well nurtured, being able to proliferate depending on the topography of the cavity [60].

Many of these nasal-resident microorganisms reach the nasal cavity from the soil or through the food used to feed the rodents. It is well acknowledged that several yeasts are ubiquitous in the environment and can be easily isolated from the soil or some plants where the fungi survive, especially when the environment is humid [61]. On the other hand, the high density of animals in more sophisticated settlements favors the dissemination of yeasts and other pathogens by frequent animal-to-animal contact and the use of contaminated water and stored manure. Both circumstances have been already highlighted by Graham et al. [4] while studying the infection of guinea pigs with *Campylobacter jejuni* in rural Ecuador. As we have shown here, guinea pigs raised intensively in cages and fed with a mixture of fodder and concentrates are colonized by more yeast species/strains in their nasal mucosa than their counterparts raised in ground pits and fed only with natural fodder. This very preliminary observation deserves to be addressed in more depth in the future.

Among the limitations of our work, the most obvious is the limited scope of the study, in geographical terms. This circumstance obliges us to consider the presented results very carefully, and to be cautious in order to not extrapolate them to other regions. Similarly, since the study was transversal, we did not consider fluctuations during different periods of the year. Finally, since we followed a classical microbiological approach—using culture methods—we did not investigate the actual diversity of the mycobiome residing in the nasal cavity of guinea pigs. In order to shed light on the actual composition and structure of this microbial community, metagenomic techniques should be used.

Nevertheless, and to the best of our knowledge, this is the first attempt to describe the diversity of culturable yeasts found in the nasal mucosa of such an important animal resource in the Andean region of Ecuador. Furthermore, our results emphasize the threat posed to human health by some of these potentially pathogenic zoonotic yeast species.

## 5. Conclusions

In summary, we isolated potentially zoonotic yeasts from the nasal duct of guinea pigs raised as livestock in Southern Ecuador. At least 11 species, belonging to 8 genera, were identified. Furthermore, the resistance exhibited by 22 of these isolates toward four antifungal drugs of therapeutic use was also revealed. This particular aspect is of fundamental importance since several of these yeast species are considered as emergent pathogens. Considering the frequent manipulation of guinea pigs without any safety measures by breeders, distributors, sellers, and consumers, the presence of pathogenic or potentially pathogenic microorganisms in their tissues and organs represents a risk from a public health point of view.

## Figures and Tables

**Figure 1 animals-12-03449-f001:**
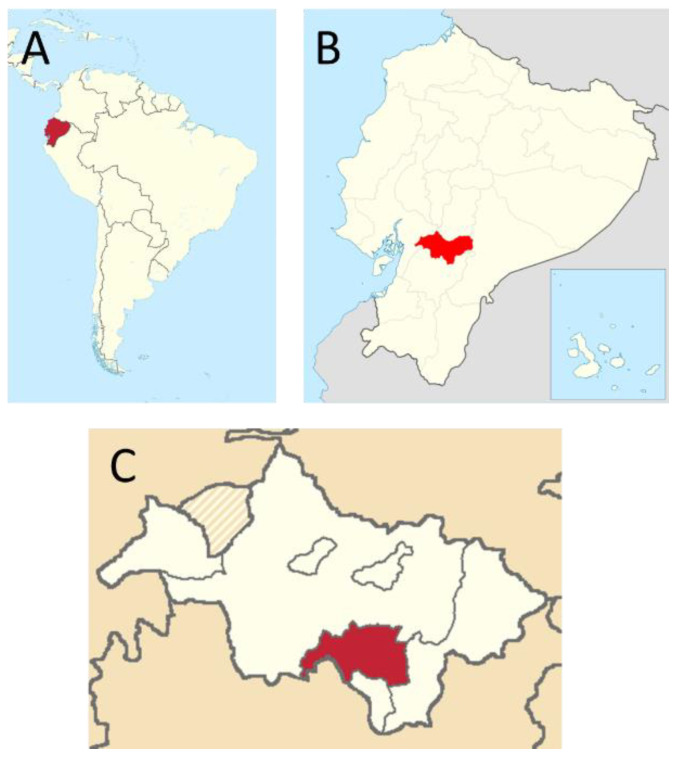
Location and map of Biblián Municipality (Cañar Province, Ecuador). (**A**) Ecuador in the South American context (in red); (**B**) Location of Cañar Province in Ecuador (in red). (**C**) Biblián Municipality (in red) (reproduced from Wikimedia Commons under Creative Commons licenses 3.0 and 4.0).

**Figure 2 animals-12-03449-f002:**
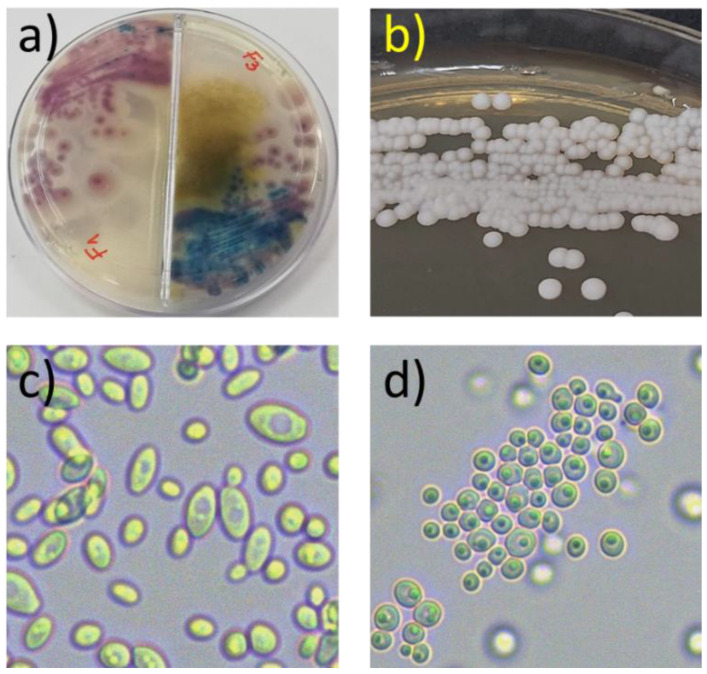
Isolation of yeasts from the nasal mucosa of guinea pigs. (**a**) Primary isolation cultures, showing pigmented colonies after 48–72 h growth at 30 °C on Hi-Crome Candida Differential Agar. (**b**) Pure isolates after re-streaking. (**c**,**d**) Microscopic view at 400× magnification of two pure isolates (isolates F2.3 and G3.1, respectively).

**Figure 3 animals-12-03449-f003:**
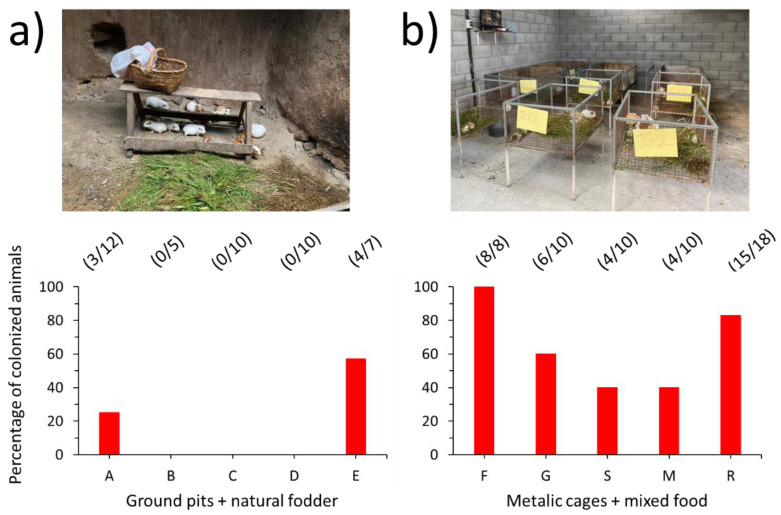
Colonization of guinea pigs’ nasal mucosa by potentially zoonotic yeasts. **Up**: Photographs of animals reared in ground pits and fed with natural fodder (**a**), or in metallic cages and fed with a mixture of fodder and cereal concentrate (**b**). **Down**: Percentage of animals colonized by yeasts in their nasal mucosa in each one of the ten farms included in this study. The number of animals actually colonized by yeasts in their nasal mucosa (as shown by cultivation in Hi-Crome Candida Differential Agar) versus the number of animals tested in each farm and of the animals is presented on top of each bar, between parentheses.

**Table 1 animals-12-03449-t001:** Molecular identification of yeast isolates by ITS1-ITS4 region sequencing and analysis.

Isolate ID	GenBank Accession Number	Closest Phylogenetic Neighbor	% Identity	% Query	Sequence Length
E4.1	ON706026	*Diutina catenulata* CBS 565	99	100	390
E6.2	ON706027	*Diutina catenulata* CBS 565	99	99	397
E6.3	ON706028	*Cryptococcus aspenensis* DS712	99.8	86	512
E7	ON706029	*Wickerhamomyces anomalus* L428/15	100	100	590
F2.1	ON706030	*Diutina catenulata* CBS 565	99.5	96	405
F2.2	ON706031	*Wickerhamomyces anomalus* L428/15	100	100	591
F2.3	ON706032	*Pichia kluyveri* E225	99.8	97	434
F2.4	ON706033	*Candida parapsilosis* LMICRO180	100	100	500
F4.1	ON706034	*Wickerhamomyces anomalus* CBS 113	100	100	590
F4.2	ON706035	*Diutina catenulata* CBS 565	99	100	398
F5.3	ON706036	*Galactomyces geotrichum* SPPRISTMF1	98	100	350
F6	ON706037	*Candida parapsilosis* IFM 63564	99.8	100	490
F9.1	ON706038	*Wickerhamomyces anomalus* CBS:261	100	100	592
G2.1	ON706039	*Wickerhamomyces anomalus* CBS 5759	99.8	100	590
G3.1	ON706040	*Wickerhamomyces anomalus* T12	100	100	422
G5.1	ON706041	(*Candida*) *railenensis* R97308	99.8	100	600
G6.2	ON706042	*Wickerhamomyces anomalus* 3Y66	100	100	580
G6.3	ON706043	(*Candida*) *railenensis* R97308	99.8	100	600
G7.2	ON706044	*Wickerhamomyces anomalus* SSL32	100	100	420
G8.1	ON706045	*Wickerhamomyces anomalus* SSL32	100	100	422
G8.2	ON706046	*Pichia fermentans* strain E224	100	100	424
G9.1	ON706047	*Wickerhamomyces anomalus* TTG-100	100	100	425
G9.2	ON706048	(*Candida*) *railenensis* R97308	99.8	100	602
G9.4	ON706049	*Pichia kluyveri* E225	99.8	100	422
MC.2	ON706050	*Meyerozyma caribbica* E12 + 4	99	98	590
MC.3	ON706051	*Meyerozyma* (*Candida*) *carpophila* FF3	98.8	99	590
MC.10	ON706052	*Meyerozyma caribbica* CBS 9966	99	99	595
PCE.3	ON706054	*Rhodotorula glutinis* H1	99.5	97	592
PCA8	ON706053	*Diutina catenulata* CBS 565	99.5	100	372
SP.8	ON706055	*Meyerozyma caribbica* CBS 9966	99.8	99	582

**Table 2 animals-12-03449-t002:** Antifungal susceptibility of yeast isolates obtained from the nasal mucosa of guinea pigs.

Isolate ID	Molecular Identification	Antifungals			
		FLU-25 *	ITC-10	VO-1	CAS-5
ATCC90028	*Candida albicans*	28	12	29	20
E4.1	*Diutina catenulata*	19	12	21	11
E7	*Wickerhamomyces anomalus*	26	NGI	18	24
F2.1	*Diutina catenulata*	17	17	31	NGI
F2.2	*Wickerhamomyces anomalus*	26	NGI	22	23
F2.3	*Pichia kluyveri*	NGI	NGI	NGI	23
F4.1	*Wickerhamomyces anomalus*	22	10	17	22
F4.2	*Diutina catenulata*	15	15	11	27
G2.1	*Wickerhamomyces anomalus*	28	12	22	23
G3.1	*Wickerhamomyces anomalus*	32	NGI	21	24
G6.2	*Wickerhamomyces anomalus*	26	NGI	14	25
G7.2	*Wickerhamomyces anomalus*	31	NGI	20	22
G8.1	*Wickerhamomyces anomalus*	27	12	30	23
G8.2	*Pichia fermentans*	NGI	NGI	NGI	23
G9.1	*Wickerhamomyces anomalus*	29	NGI	20	22
G9.2	(*Candida*) *railenensis*	15	7	10	30
G9.4	*Pichia kluyveri*	NGI	NGI	NGI	21
MC10	*Meyerozyma caribbica*	27	NGI	39	NGI
MC2	*Meyerozyma caribbica*	30	NGI	31	NGI
MC3	*Meyerozyma carpophila*	24	NGI	31	NGI
PCA.8	*Diutina catenulata*	16	11	22	10
SP8	*Meyerozyma caribbica*	29	NGI	30	12
PCE3	*Rhodotorula glutinis*	NGI	NGI	ND	ND

Pure isolates were spread on the surface of Mueller–Hinton Agar plates supplemented with 2% glucose and 0.5 µg mL^−1^ methylene blue dye. The test was performed according to the Clinical and Laboratory Standards Institute (CLSI) approved protocol [25], using paper discs loaded with the respective antifungal drug. * The diameter of the zone of inhibition for each isolate was measured after incubation for 24 h at 30 °C and is expressed in mm. NGI: No growth inhibition (inhibition halo absent); ND: Not determined. FLU-25: Fluconazole (25 mcg); ITC-50: Itraconazole (50 μg); VO-1: Voriconazole (1 μg); CAS-5: Caspofungin (5 μg).

## Data Availability

Not applicable.

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
