# Peer review of "Role of Guinea Pigs (Cavia porcellus) Raised as Livestock in Ecuadorian Andes as Reservoirs of Zoonotic Yeasts"

_animals, 2022, doi:10.3390/ani12243449_

Round 1

Reviewer 1 Report

The study presents sound methodology to identify fungi. However, the authors fail to prove the threat posed by potentially zoonotic yeast to the final consumers and animal health. For instance should describe how is the handling of the products from the farm to the consumption also how the presence of contamination in that particular farms can represent a risk for the animal health. Correct these statements on the conclusion.

Line 60: substitute the words "Yet" by Additionally and "owing" by towards. 

Author Response

The study presents sound methodology to identify fungi. However, the authors fail to prove the threat posed by potentially zoonotic yeast to the final consumers and animal health. For instance should describe how is the handling of the products from the farm to the consumption also how the presence of contamination in that particular farms can represent a risk for the animal health. Correct these statements on the conclusion.

A: Thank you very much for the comment. Following your suggestion, we included the following statement: “Considering that i) guinea pigs are among the most important sources of animal protein for human consumption in the Andean region; ii) these rodents are frequently handled by breeders, distributors, sellers and consumers and iii) almost no safety measures are taken during breeding, slaughter, sale and preparation, a better understanding of their role as potential reservoirs of opportunistic pathogens is necessary.” (please, see lines 217 to 222, in the revised version)

We also added a small comment, in the Conclusion section, as follows: “Considering the frequent manipulation of these animals, by breeders, distributors, sellers and consumers, without any safety measures, the presence of pathogenic or potentially pathogenic microorganisms in their tissues and organs could represent a risk from a public health point of view.” (please, see lines 351 to 354)

Line 60: substitute the words "Yet" by Additionally and "owing" by towards.

A: Thank you. We modified this sentence as follows: “Yet, in the last 20 years a few reports warned about the identification of guinea pigs as either accidental hosts or as bona fide reservoirs for human pathogens.” (please, see lines 61 to 62)

Reviewer 2 Report

The manuscript entitled " Role of guinea pigs (Cavia porcellus) raised as livestock in Ecuadorian Andes as reservoirs of zoonotic yeasts " is well organized. The idea and design of this study is good although it is not clear. However, I suggest the authors should address the following major comments for further consideration of the manuscript for publication.
I suggested improve the background of the study.
The introduction must be improved with more relevant information about the core plot of the manuscript.

The authors should clearly define the protocol use in this experiment and this is applied to the results part.

Discussion: Revise the manuscript with strong justification for background.
Conclusion: write the conclusion again and be more concise to the major findings and suggestions.

I find some formal errors in the manuscript, above all it is necessary to check the correctness of the literary references in the text according to the list of references (dates of publication) and also the correctness of writing the references in the text (dots, commas, spaces...). Not all abbreviations are explained in the text.

It is general to italicize family, genus, and species, but not name, of viruses on their taxonomy.

Grammar, punctuation, sentence structure, use of past/present tense and the use of plurals within the manuscript are quite consistent. Thus, it may be to the authors benefit to make use of a professional editing service to perfect the manuscript.

Author Response

The authors thank to the reviewer for the constructive and valuable comments and suggestions. Here are our answers to specific comments or questions:

I suggested improve the background of the study.

A: Thank you very much for the suggestion. In fact, we did mention, in the Introduction, that our initial aim “…was to isolate and identify Candida spp. species colonizing the nasal mucosa of healthy guinea pigs, raised in Southern Ecuador as livestock, and to test their susceptibility against several antifungal drugs”. This was our initial expectation, based on the frequent detection of such type of microorganisms in this particular habitat (i.e. the nasal mucosa) (please, see lines 229-230, and 313-319). We also explain, in the Discussion section, that the isolation of so many different yeast species was a surprise to us. We thus thought that it was relevant to publish this serendipitous discovery, as a first and preliminary observation, to pave a new way for future studies.

The introduction must be improved with more relevant information about the core plot of the manuscript.

A: Thank you very much for your suggestion. We believe we have developed an argument based on i) the role of farm animals as reservoirs of human pathogens; ii) the frequent infection of guinea pigs (farm animals in the Andean countries) by human pathogens; iii) the importance of some yeast species as emergent pathogens; iv) the colonization of domestic/wild animals by this type of opportunistic pathogens, and, finally; v) the absence of any studies concerning this scientific gap (i.e. the case of guinea pigs raised for human consumption). It is our humble opinion that this defines the core pot of our manuscript.

The authors should clearly define the protocol use in this experiment and this is applied to the results part.

A: We are sorry: we do not see to which protocol the reviewer is referring at…To the best of our knowledge, all experimental details are clearly provided in the Material and Methods section.

Discussion: Revise the manuscript with strong justification for background.

A: This general comment was already addressed before.

Conclusion: write the conclusion again and be more concise to the major findings and suggestions.

A: The conclusion was modified, following all the reviewers’ suggestions. It reads now as follows: “In summary, we isolated potentially zoonotic yeasts from the nasal duct of guinea pigs, raised as livestock in Southern Ecuador. At least 11 species, belonging to eight gene-ra, were identified. Furthermore, the resistance exhibited by 22 of these isolates toward four antifungal drugs of therapeutic use was also revealed. This particular aspect is of fundamental importance, since several of these yeast species are considered as emergent pathogens. Considering the frequent manipulation of guinea pigs, without any safety measures, by breeders, distributors, sellers and consumers, the presence of pathogenic or potentially pathogenic microorganisms in their tissues and organs represents a risk from a public health point of view.”

I find some formal errors in the manuscript, above all it is necessary to check the correctness of the literary references in the text according to the list of references (dates of publication) and also the correctness of writing the references in the text (dots, commas, spaces...). Not all abbreviations are explained in the text.

A: We corrected some minor errors (see, for instance, lines 65, 251, 259, 261, 263, 270, 281 and 284). Concerning the rest of the formal errors mentioned by the reviewer, it would have been nice to receive more specific comments, in order to be able to correct them. However, we did our best in order to correct some mistakes, which are highlighted in red in the revised version.

The following abbreviations were defined: European Committee on Antimicrobial Susceptibility Testing (EUCAST), Clinical & Laboratory Standards Institute (CLSI), Food and Agriculture Organization (FAO) of the United Nations.

The following abbreviations were not defined (as they are familiar for the general reader): PCR, DNA, rDNA, MIC, BLAST, RT.

It is general to italicize family, genus, and species, but not name, of viruses on their taxonomy.

A: Thank you for the comment. However, as far as we can tell, we did not include such an italicized virus name on our manuscript.

Grammar, punctuation, sentence structure, use of past/present tense and the use of plurals within the manuscript are quite consistent. Thus, it may be to the authors benefit to make use of a professional editing service to perfect the manuscript.

A: Thank you very much!

Reviewer 3 Report

The manuscript by Buela et al. is a survey on role of Cavia porcellus as reservoir of potentially zoonotic yeasts in Ecuadorian Andes (as guinea pigs are one of the main ingredients of traditional food).

In my opinion, this manuscript must be resubmitted only after very substantial revisions.

Some major and minor comments are listed in the attached file.

Author Response

The authors thank Reviewer 3 (in particular) for the constructive and valuable comments and specific suggestions. Here are our answers to specific comments or questions:

- according to the Authors in “Abstract” (line 36) and “Conclusions” (line 349) five genera were identified; why in “Results” (line 178) genera are seven?

A: Thank you for the comment. We made several numeric mistakes, and we deeply apologize for that. The errors were corrected, after careful revision of the information provided in the Results section (and, also, after careful revision of our students’ lab notebooks). As you will see (in the Abstract, the Results and the Conclusion sections) the strains were preliminarily identified as members of eleven species, belonging to eight genera.  

- Lines 89-90: the coordinates indicated by the Authors correspond to a Brazilian place...

A: The referee is right. This was corrected to 2°42′36″ S, 78°52′48″ W, which is the actual location of the Biblián Cantón.

- In “Materials and methods” (lines 120-121) 22 pure isolates were selected for MHA test; what criteria were used in this selection?

A: In fact, since this study was performed during the second year of the COVID-19 pandemic, some isolates could not be reactivated and were lost, due to dehydration of some plates (after several weeks of limited mobility). This was the main reason why we performed the MHA test with only 22 isolates. 

- In “Results” (Line 149) yeasts are reported in 39 out of 97 animals tested (40.2%), but in Fig 3b, (lines 169-172) the sum of positive animals versus animals tested in all farms is 44/100.

A: Thank you for the observation. We carefully checked the results (by directly consulting the lab notebooks of our students) and corrected to 44 positive animals of 100 included in the survey.

- In Table 1 are reported five isolates belonging to C. railanensis and C. parapsilosis, but in “Discussion” (line 297) the isolates are four.

A: The referee is right. We corrected this mistake in line 295 of the Discussion section.

- Fig 4 and Table 2: why is E.S.2 (Fig. 4) not present in Table 2?

A: Thank you very much for this particular comment. In fact, it is actually strain “E.6.2”. This strain is not presented in Table 2 because we did not test all of the same (four) antifungals used to build this table. That is why we preferred to exclude this (and a few other) strains from this table.  Considering your comment below, and in order not to confuse our readers, Figure 4 was deleted from this revised version.

Minor comments

In my opinion, there are several redundancies: in Table 2 and Fig. 4, e.g., the description of the methods in the captions is redundant.

A: Thank you for the observation. Since Figure 4 was presented to the readers for illustration purposes only, we decided to remove it from the revised version of our manuscript.

Line 99: IN farm animals

A: Corrected (please, see line 99 in the revised version)

Line 102: I think that “written consent” in sufficient

A: Corrected (please see line 102 in the revised version)

Line 239: add appropriate references

A: Thank you. We added the following reference: 30.   Deorukhkar, S.C.; Saini, S.; Mathew, S. Non-albicans Candida infection: An emerging threat. Interdiscip Perspect Infect Dis 2014, 615958. doi: 10.1155/2014/615958

Line 274: add appropriate references

A: Thank you. We added the following reference: 47.     Chen, X-F; Zhang, W.; Fan, X.; Hou, X.; Liu, X-Y.; Huang, J-J.; Kang, W.; Zhang, G.; Zhang, H.; Yang, W-H.; Li, Y-X.; Wang, J-W.; Guo, D-W.; Sun, Z-Y.; Chen, Z-J; Zou, L-G.; Du, X-F.; Pan, Y-H.; Li, B.; He, H.; Xu, Y-C. Antifungal susceptibility profiles and resistance mechanisms of clinical Diutina catenulata isolates with high MIC values. Front Cell Infect Microbiol 2021, 11,739496. doi: 10.3389/fcimb.2021.739496

Line 305: UPMOST

A: Corrected (please, see line 303 in the revised version)

Round 2

Reviewer 3 Report

I think that the manuscript has been sufficiently improved to warrant publication in Animals.